# Plasticity and evolutionary convergence in the locomotor skeleton of Greater Antillean *Anolis* lizards

**Nathalie Feiner[1]\*, Illiam SC Jackson[1†], Kirke L Munch[2], Reinder Radersma[1‡], Tobias Uller[1]**

[1]Department of Biology, Lund University, Lund, Sweden; [2]School of Biological Sciences, University of Tasmania, Hobart TAS, Australia

**Abstract** Plasticity can put evolution on repeat if development causes species to generate similar morphologies in similar environments. *Anolis* lizards offer the opportunity to put this role of developmental plasticity to the test. Following colonization of the four Greater Antillean islands, *Anolis* lizards independently and repeatedly evolved six ecomorphs adapted to manoeuvring different microhabitats. By quantifying the morphology of the locomotor skeleton of 95 species, we demonstrate that ecomorphs on different islands have diverged along similar trajectories. However, microhabitat-induced morphological plasticity differed between species and did not consistently improve individual locomotor performance. Consistent with this decoupling between morphological plasticity and locomotor performance, highly plastic features did not show greater evolvability, and plastic responses to microhabitat were poorly aligned with evolutionary divergence between ecomorphs. The locomotor skeleton of *Anolis* may have evolved within a subset of possible morphologies that are highly accessible through genetic change, enabling adaptive convergence independently of plasticity.

**\*For correspondence:**
nathalie.feiner@biol.lu.se

**Present address:** [†]College of Natural Sciences, University of Texas at Austin, Austin, United States; [‡]Biometris, Wageningen University & Research, Wageningen, Netherlands

**Competing interests:** The authors declare that no competing interests exist.

## Introduction

Darwin famously described evolution by natural selection as a source of 'endless forms'. While the diversity of life may indeed seem endless, the many examples of convergent or parallel evolution reveal that evolution commonly is on repeat. Both the developmental biology and the ecology of organisms contribute to such instances of repeated evolution. The mechanisms of development make some morphologies arise frequently and others rarely (i.e., developmental bias; *Uller et al., 2018*), while the ecology of the population makes some of the morphologies that do arise consistently fitter than others. These explanations have often been pitted against each other ('constraint' *versus* 'selection'), but this is a false dichotomy. For example, the striking parallelism of cichlid fish in Lake Malawi and Lake Tanganyika is jointly determined by the similarity in the ecology, and hence selective pressures, of the lakes, and the shared developmental biology of the fish that provided natural selection with similar morphologies (*Kocher et al., 1993*).

*Anolis* lizards are one of the iconic examples of evolution on repeat. Following colonization of the Greater Antillean islands (Cuba, Hispaniola, Jamaica and Puerto Rico) some 50 mya, lizards have diversified by exploiting a variety of habitats, including tree trunks, twigs and bushes (*Rand and Williams, 1969*; *Williams, 1972*; *Figure 1A*). While the role of natural selection in the adaptive radiation of *Anolis* is undisputed, it remains poorly understood how developmental bias has shaped their phenotypic diversification and convergence. One hypothesis posits that plastic responses to the microhabitat contributed to, and perhaps facilitated, the evolution of similar morphologies (i.e., 'ecomorphs') on different islands (*Losos et al., 2000*; *Losos et al., 2001*; *West-Eberhard, 2003*). Plasticity is conducive to repeated evolution when the developmental mechanisms that enable

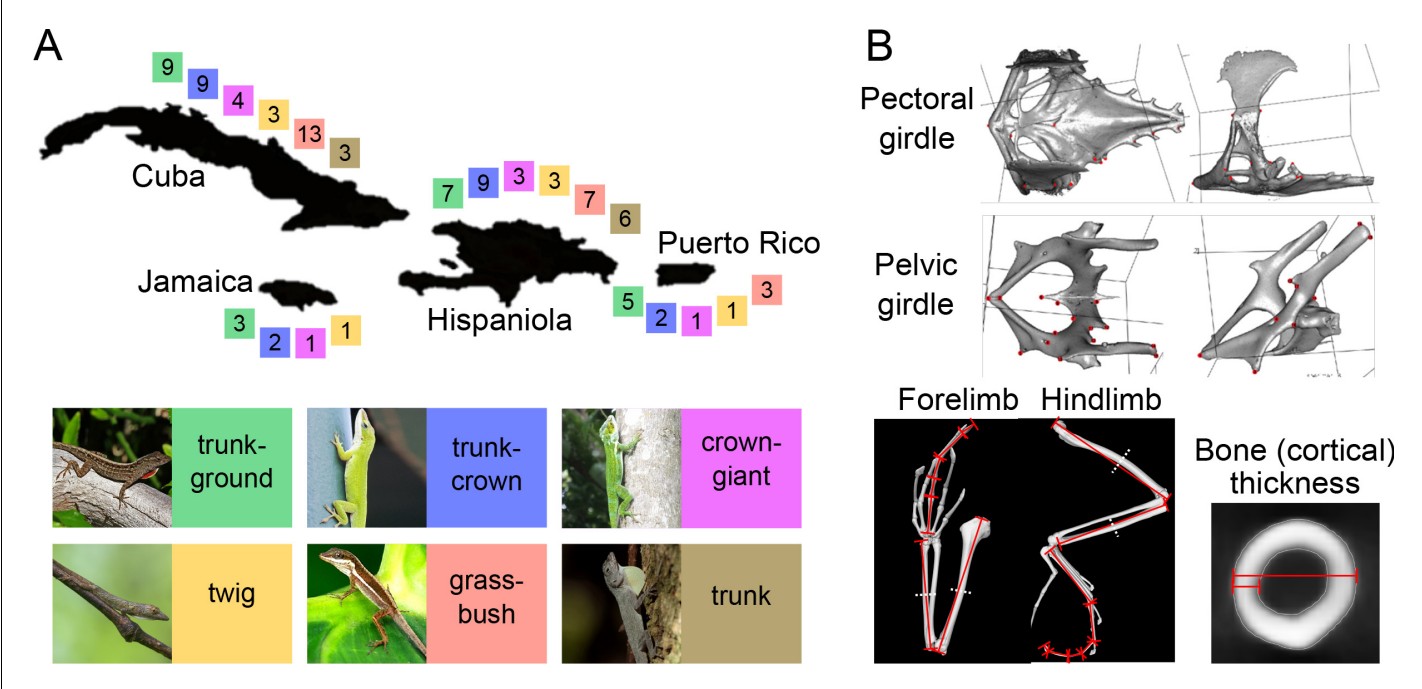

**Figure 1.** Distribution of ecomorphs on the four main Greater Antillean islands and elements of the morphological dataset capturing variation in the locomotor skeleton. (**A**) The four islands of the Greater Antilles, Cuba, Jamaica, Hispaniola and Puerto Rico, each inhabit a set of independently evolved ecomorphs (*Poe et al., 2017*). Colour-coded squares show the number of species that were used in this study. Note that Jamaica and Puerto Rico do not possess the full set of six ecomorphs, but lack one and two ecomorphs, respectively. Thumbnail pictures depict representative species for each ecomorph: *A. sagrei* (trunk-ground), *A. carolinensis* (trunk-crown), *A. luteogularis* (crown-giant), *A. occultus* (twig), *A. pulchellus* (grass-bush), *A. distichus* (trunk). (**B**) Morphological variation was quantified on the basis of 18 3D landmarks on each of the pectoral and pelvic girdles (for more information, see *Supplementary file 1A*), the length of individual bones of both fore- and hindlimb, and bone thickness and bone cortical thickness of long bones. (**A**). thumbnail permissions: 'Image of *A. sagrei* (trunk-ground), courtesy of Brown, 2013, retrieved from https://commons.wikimedia.org/wiki/File:Cuban_Brown_Anole_(Anolis_sagrei)_(8592690316).jpg on 07/22/2020, distributed under the terms of the CC-BY 2.0 license.'.

individuals to accommodate new conditions are evolutionarily conserved. This is particularly relevant to the locomotor skeleton (i.e., limbs, shoulder and hip girdles) given that bone growth responds to mechanical stress (*West-Eberhard, 2003*; *Hall, 2005*), that microhabitat is known to affect limb growth in lizards (*Losos et al., 2000*; *Kolbe and Losos, 2005*; *Downes and Hoefer, 2007*; *Langford et al., 2014*), and the well-documented ability of *Anolis* populations to adaptively evolve limb length to fit their structural habitat (*Schoener et al., 2004*; *Stuart et al., 2014*). Thus, the locomotor skeleton of *Anolis* species could be prone to evolve similar morphologies on different islands because the mechanical stress imposed by each habitat makes some functional solutions (i.e., fit morphologies) consistently more accessible to selection than others.

It has previously been suggested that the repeated adaptive match between microhabitat and limb length represents an example of such 'plasticity-led' evolution (*Losos et al., 2000*; *Losos et al., 2001*; *West-Eberhard, 2003*). However, since limb length can only vary in one dimension (i.e., longer or shorter), it provides limited information on the alignment between plasticity and evolutionary divergence. Multivariate data are more suitable for sorting between alternative adaptive solutions to the same problem (i.e., many-to-one mapping between morphology and performance). Such data make it possible to assess if traits that are highly plastic are also highly evolvable, and if plastic responses are well aligned with adaptive divergence (*Radersma et al., 2020*).

To assess the extent to which morphological plasticity is likely to have shaped the adaptive radiation of *Anolis* lizards, we quantified the shape of the pectoral (shoulder) and pelvic (hip) girdles, the length of individual limb bones and the thickness of the long bones for 95 species of Greater Antillean *Anolis*. Using these data, we firstly assessed to what extent the morphological divergence of pairs of ecomorphs are parallel on the four main islands. Secondly, we tested if the features of the locomotor skeleton that respond to mechanical stress are particularly evolvable and, thirdly, if morphological divergence between those ecomorphs is aligned with the morphological plasticity induced directly by the microhabitat. Both of these associations are expected if plasticity were to contribute to the repeated evolution of similar morphologies. However, environmentally induced morphologies only shape evolution if the changes have functional consequences. Finally, we therefore assessed if evolutionary divergent and phenotypically plastic features of the locomotor skeleton predict perching behavior and locomotor performance in species adapted to different microhabitats.

## Results

Using micro-CT scans of museum specimens, we quantified the morphology of the locomotor skeleton of 259 males from 95 Greater Antillean *Anolis* species that have been assigned to one of six ecomorph categories (*Losos, 2009*; *Nicholson et al., 2012*; *Figure 1A*). We acknowledge that the ecomorph classification does not always rely on well-documented habitat use (*Poe and Anderson, 2019*), but there is a very close concordance between classification based on habitat use (*Nicholson et al., 2012*) and classification that also includes gross morphology, such as the total length of limbs, head size and tail length (*Losos, 2009*). We quantified shape using 18 3D landmarks each on the pectoral and the pelvic girdle (partially adapted from *Tinius and Russell, 2014*; *Tinius et al., 2018*; *Supplementary file 1A*), and measured the length of 15 limb elements of fore- and hindlimb, as well as the cortical thickness and bone diameter in virtual cross-sections of the four long bones (*Figure 1B*). We used the centroid size of the pelvic girdle as a proxy for body size. This resulted in a morphological dataset consisting of 132 features or traits that were scaled to allow vector-based analyses on combined shape and univariate data (*Adams and Collyer, 2019*), following a similar approach by *Stuart et al., 2017*; see Materials and methods). While this approach means that differences between, for example, ecomorphs are not interpretable in terms of shape, the data capture the overall morphology of the locomotor skeleton, which is the primary focus of this study. However, for completeness, we also report the main results for girdles and limbs analysed separately. Note that this data set is very different from the traits (e.g., total limb, head and tail lengths) that form the basis for most inference on *Anolis* adaptive radiation and convergence (e.g., *Mahler et al., 2013*). Thus, our study neither replicates these previous studies, nor does it follow a priori that the locomotor skeleton will be as convergent as gross morphology.

### Convergence of ecomorphs on the Greater Antillean islands is moderate, but consistent

Principal component analyses of the total dataset showed that ecomorphs of different islands group roughly in the same area of morphospace, albeit with substantial overlap. Crown-giant, twig and grass-bush ecomorphs tended to occupy private areas of morphospace, while all other ecomorphs formed a single cloud (*Figure 2*; detailed results in *Figure 2—figure supplement 1*).

Matching the differences *between* ecomorphs, another hallmark of evolution on repeat are convergences *within* one ecomorph (i.e., convergence of morphology of lizards found in a particular microhabitat). We adopted a methodology using a simple metric of convergence in a phylogenetic framework, the Wheatsheaf index (*Arbuckle et al., 2014*), to assess if levels of convergence for each

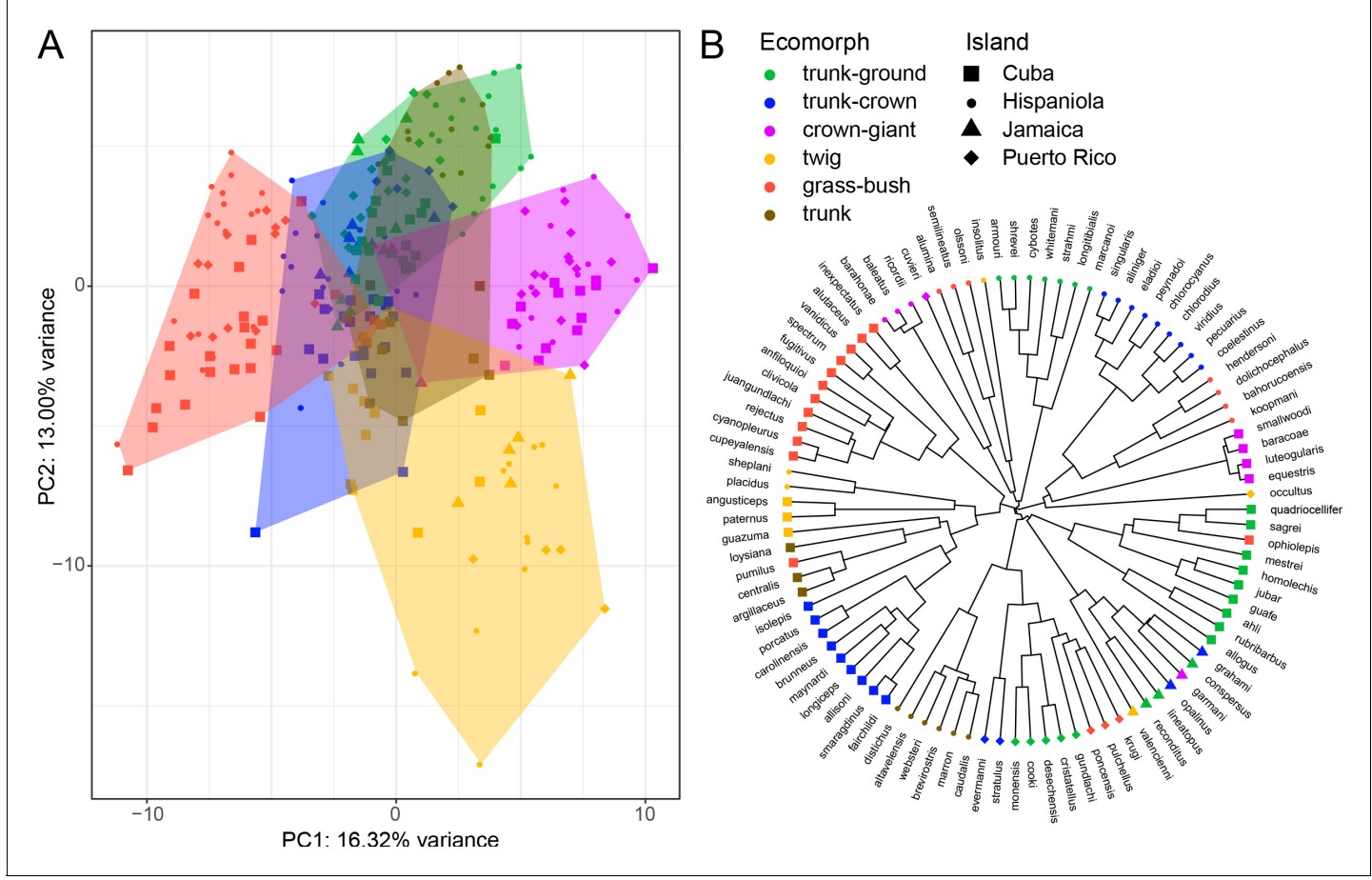

**Figure 2.** Principal component analysis of 259 individual males from 95 *Anolis* species and their phylogenetic relationships. (**A**) The first and second principal components on the level of individual explain 16.32% and 13.00% of the variance, respectively. The features that load most strongly on PC1 are the shape of the pelvic girdle and humerus length, and features loading strongly on PC2 are the shape of the pectoral girdle and bone cortical thickness (*Supplementary file 1L*). (**B**) Phylogenetic tree based on *Poe et al., 2017* with the associated island of origin and ecomorph classification (*Losos, 2009*; *Nicholson et al., 2012*) per species. For more detailed results of species-level and phylogenetic principal component analysis, see *Figure 2—figure supplement 1*.

The online version of this article includes the following figure supplement(s) for figure 2:

**Figure supplement 1.** Principal component analysis of 259 individual males from 95 *Anolis* species in a phylogenetic context.

ecomorph are higher than expected by chance given the phylogenetic relatedness. Across all traits, convergence was weak and not statistically significant (p-values≥0.13), but crown-giants showed stronger convergence than other ecomorphs (*Supplementary file 1B*). However, convergence was greater than expected by chance for some traits, and these 'signature traits' (*Supplementary file 1C*) were distinct for each ecomorph (except for one shape feature of the pelvic girdle which is characteristic for both grass-bush and trunk ecomorphs). For example, the most convergent traits for grass-bush lizards were associated with the shape of the pelvic girdle, while the most convergent traits of trunk-crown lizards were associated with the shape of the pectoral girdle (*Supplementary file 1C*). Crown-giants had the highest number of signature traits (N = 15), while trunk ecomorphs had the lowest (N = 3, *Supplementary file 1C*). These results show that none of the traits of the locomotor skeleton are consistently evolutionarily labile and repeatedly adapted to different microhabitats.

## Phenotypic trajectories between ecomorphs across islands are more aligned than expected by chance

We assessed the extent to which the differences between ecomorph pairs were consistent on the four islands following the logic of phenotypic trajectory analysis (*Adams and Collyer, 2009*; *Collyer et al., 2015*; see also *Stuart et al., 2017*). Evolution can be considered highly repeatable if a vector that describes the morphological difference between a pair of ecomorphs on one island is parallel to and of the same magnitude as the same vector for another island. For pairs of ecomorphs on a given island, we performed t-tests for each trait, which resulted in a vector of 132 *t*-values that describes the trajectory between ecomorphs in multivariate space (*Figure 3*). The angle (Θ) and the difference in length (ΔL) between two vectors of different islands is then calculated to assess how parallel they are, and significant deviations from parallelism and random alignment were assessed using randomization and permutation.

Across islands, divergence in the locomotor skeleton between any pair of ecomorphs were consistently better aligned than expected by chance, with a mean angle of 54.44° (standard deviation: 12.57°; *Figure 4A*, *Supplementary file 1D*). Only one out of 53 comparisons did not significantly deviate from orthogonal (90°; *Figure 4B*, *Supplementary file 1D*). However, only in five instances was the divergence of ecomorph pairs sufficiently similar for a pair of islands to be indistinguishable (angles not significantly different from 0°, *Figure 4B*, *Supplementary file 1D*). When considering the

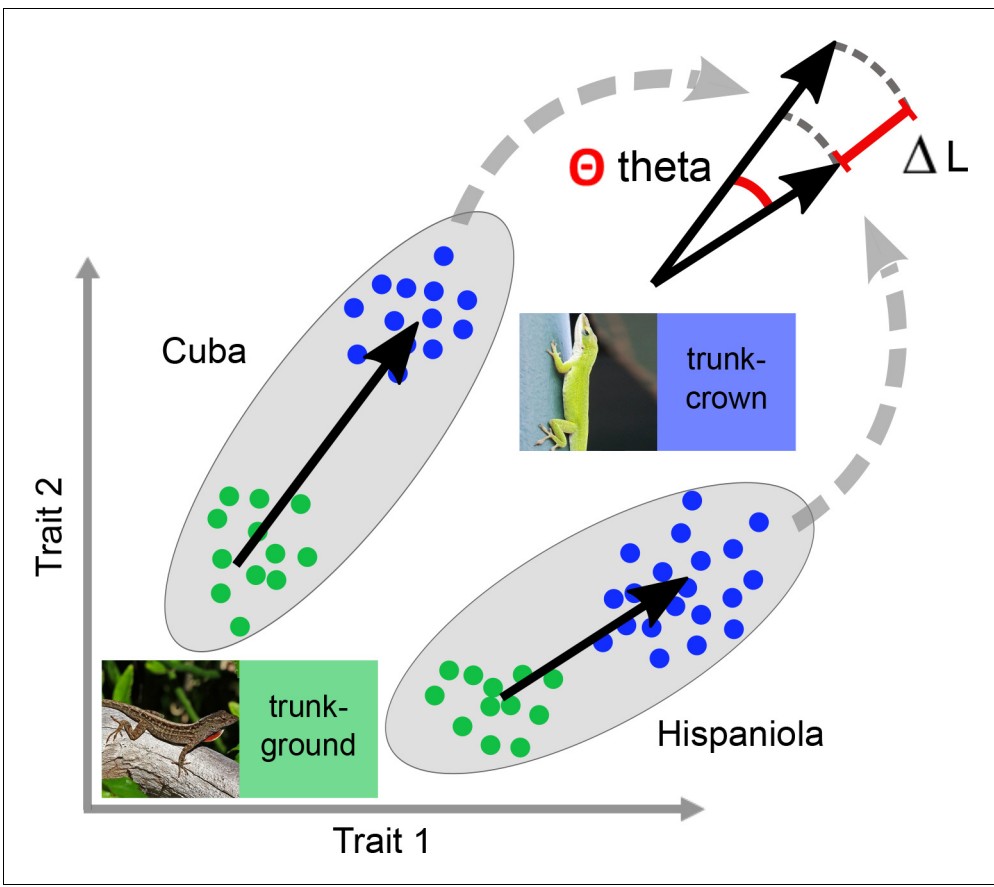

**Figure 3.** Schematic for assessing how well divergence between ecomorph pairs are aligned between islands. The rationale of assessing the degree of alignment uses a vector-based approach in multivariate space, but is depicted for a schematic 2-trait space here to demonstrate the logic. The alignment of island pairs was quantified as the angle theta (Θ) between the two vectors that describe the trajectories between sets of two ecomorphs in multivariate space. An angle of zero implies that the two vectors are parallel. The same logic applies to the difference between the lengths (ΔL) of the vectors. The trajectory between ecomorphs was calculated based on t-test statistics per trait (*Adams and Collyer, 2009*; see also *Stuart et al., 2017*). For thumbnail permissions, see *Figure 1*.

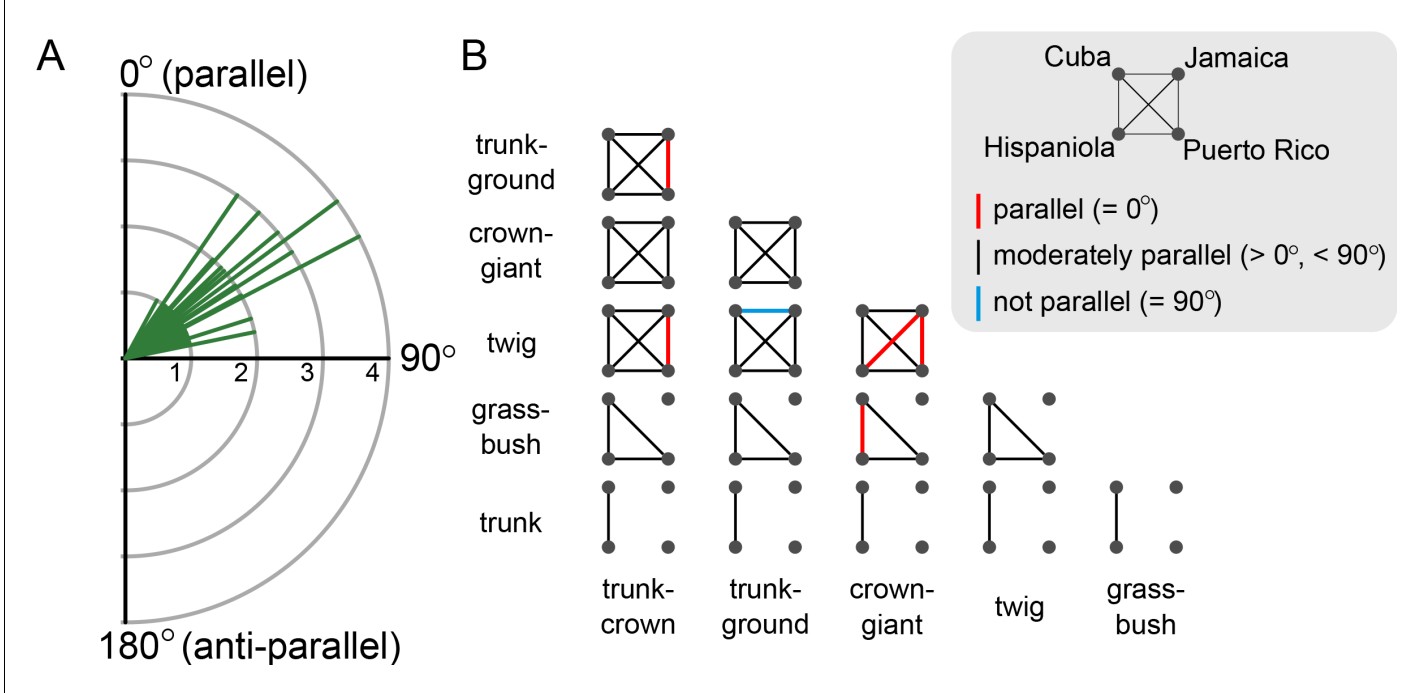

**Figure 4.** Alignment of the morphological difference between pairs of ecomorphs on the four Greater Antillean islands. (**A**) A frequency plot of all 53 quartet comparisons (pairs of ecomorphs on island pairs) shows that all angles Θ are between 0° and 90° (mean: 54.44°; standard deviation: 12.57°). (**B**) A matrix of ecomorph pairs is used to visualize the significance level of parallelism between island pairs. Each line represents a pair of islands and the colour signifies if the angle is indistinguishable from 0° (red), significantly larger than 0°, but smaller than 90° (grey), or indistinguishable from 90° (blue). Significance levels were assessed using randomization and permutation tests (see Materials and methods).

The online version of this article includes the following figure supplement(s) for figure 4:

**Figure supplement 1.** Alignment of the morphological difference between pairs of ecomorphs on the four Greater Antillean islands separately for individual aspects of the locomotor apparatus.

**Figure supplement 2.** Rearing habitat used in locomotion-induced plasticity experiment.

difference in magnitude of ecomorph divergence between islands (i.e., ΔL between two vectors), we find that 14 out of 53 vector pairs are indistinguishable in length (*Supplementary file 1E*). The results were consistent when pectoral girdle, pelvic girdle, limb length and bone thickness were analysed separately (*Figure 4—figure supplement 1*), and with an alternative method that relies on comparisons of multivariate means instead of t-tests (*Collyer and Adams, 2007*). Overall, we conclude that the pattern of moderate but consistent morphological divergence between pairs of ecomorphs on the Greater Antilles is not driven by a particular anatomical category, but distributed across the locomotor skeleton.

## Evolutionary divergence between ecomorphs is poorly aligned with habitat-induced morphological plasticity

Next, we addressed if traits that show evolutionary divergence between pairs of ecomorphs are also the traits that are most responsive to mechanical stress imposed by the microhabitat, a situation that should apply if plasticity was an evolutionarily important source of developmental bias. To test this hypothesis, we raised 45 male lizards of a trunk-crown (*Anolis carolinensis*, adapted to climbing) and 71 male lizards of a trunk-ground (*A. sagrei*, adapted to running) species on broad *versus* narrow surfaces until they reached adulthood (*Figure 4—figure supplement 2*). The differences in the structural habitats promoted different styles of locomotion and perching behaviour (*Feiner et al., 2020*). Both species belong to the Cuban clade and, therefore, the appropriate evolutionary divergence to compare against is the contrast between the respective ecomorphs in this clade on Cuba (trunk-ground [$N_{Individuals}$ = 20, $N_{Species}$ = 9] *versus* trunk-crown [$N_{Individuals}$ = 20, $N_{Species}$ = 9]).

The morphology of both species responded to the microhabitat treatment. As expected, the magnitude of environmentally induced morphological change was substantially smaller than the evolutionary divergence between the corresponding ecomorphs. The vector length was three times larger for the evolutionary divergence compared to the environmentally induced shift in morphology (*A. sagrei*$_{(broad\ vs\ narrow)}$, L = 12.05; *A. carolinensis*$_{(broad\ vs\ narrow)}$, L = 12.23; ecomorph$_{(trunk-ground\ vs\ trunk-crown)}$, L = 39.47). Among all 132 traits, we singled out 15 traits that significantly distinguish between Cuban trunk-ground and trunk-crown specimens (i.e., the most evolutionary divergent traits), and 14 and 10 traits that significantly distinguish the environmentally induced microhabitat specialists in *A. carolinensis* and *A. sagrei*, respectively (i.e., most plastic traits; see Material and methods). Notably, these traits showed limited overlap between the three runner/ climber contrasts: while *A. carolinensis* showed the greatest plastic response in the anterior part of the locomotor skeleton, the most affected traits in *A. sagrei* were found in the posterior part associated with the pelvic girdle or the hindlimb (*Supplementary file 1F*). In contrast, the 15 most evolutionary divergent traits were mostly associated with limb bones (*Supplementary file 1F*). This qualitative difference between the runner/climber axis of ecomorphs and environmentally induced phenotypes is supported by comparing angles between runner/climber vectors. Using all traits, the direction of the environmentally induced shift in multivariate space was not aligned between the two species (80.84°) and none of them was aligned with the evolutionary divergence between the two ecomorphs on Cuba (*A. carolinensis*, 81.22°; *A. sagrei*, 105.96°; *Supplementary file 1G*).

Tests of plasticity-led evolution commonly rely on traits that are known to be either environmentally responsive or adaptively divergent (*Levis and Pfennig, 2016*). We therefore repeated the analyses above using only the most plastic traits of the locomotor skeleton of each species. This demonstrated again that there was no alignment between plasticity and ecomorph divergence (*A. carolinensis*, 87.84°; *A. sagrei*, 116.67°; *Supplementary file 1G*). Similarly, a vector of the 15 traits that show the strongest evolutionary divergence between trunk-ground and trunk-crown lizards on Cuba was not significantly aligned with the plasticity vector of *A. carolinensis* (40.20°) or *A. sagrei* (125.98°; *Supplementary file 1G*).

## Morphological plasticity does not consistently increase locomotor performance

To assess how well morphology predicts functional aspects related to fitness, we performed two trials that capture perching behaviour and locomotor performance in the lizards that were raised in the two microhabitats (see Material and methods). First, we assessed perching behaviour, and second, we scored running performance on broad and narrow surfaces (see *Feiner et al., 2020*) for more details). We have previously shown that locomotor performance was enhanced in the familiar microhabitat (*Feiner et al., 2020*). Here, we demonstrate that this effect was largely uncoupled from morphological plasticity: in both species, the first two principal components of the 14 resp. 10 most plastic traits failed to explain individual variation in sprint speed (on narrow and broad surfaces) and perching behaviour, while PC2 had a minor effect on perching behaviour in *A. carolinensis* only (*Supplementary files 1H and 1I*). In both species, at least one of the first two principal component of the most evolutionarily divergent traits explained variation in locomotor performance (*Supplementary files 1H and I*).

## Discussion

The repeated evolution of similar phenotypes in different species provides outstanding opportunities to understand developmental and ecological causes of evolution. Plasticity can promote repeated evolution when the developmental mechanisms are conserved, since species will tend to generate similar phenotypes under similar ecological conditions. To test this hypothesis, we conducted a detailed analyses of the morphology of the locomotor skeleton of *Anolis* lizards in the Greater Antilles and compared this to the morphological variation induced by locomotion in different microhabitats. While the morphological divergence in the locomotor skeleton between ecomorphs occurred along similar trajectories on the four islands, the results suggest that phenotypic plasticity at best played a minor role in the repeated evolution of these morphologies.

The adaptive radiation of *Anolis* in the Greater Antilles appears to have filled the morphospace of the locomotor skeleton homogenously, without leaving gaps. While some ecomorphs are rather

distinct in their morphology (e.g., crown-giant, twig and grass-bush), others show substantial over-lap. Nevertheless, morphological divergence between pairs of ecomorphs tended to occur along similar trajectories, even if the ecomorph divergence on one island rarely was indistinguishable from that of another island (i.e., $\Theta \approx 0$). The alignment of morphological divergence was highly consistent for all ecomorph pairs across the four islands, and this result held up in separate analyses of limb bones and the pectoral and pelvic girdles. Thus, *Anolis* lizards adapted to a given microhabitat show a moderate degree of morphological similarity in their locomotor skeleton, a similarity that appears consistent across the Greater Antillean islands (see also *Tinius and Russell, 2014*; *Tinius et al., 2018*).

Several different morphologies of limbs and girdles can perform well in a given microhabitat (i.e., a many-to-one mapping of form to function; *Wainwright et al., 2005*; *Alfaro et al., 2006*; *Vanhooydonck et al., 2006*). Thus, repeated evolution is more likely when species share a developmental biology that makes some phenotypes appear readily while others are rare or even impossible. For example, *Anolis* lizards that live on tree trunks would be prone to adapt in similar ways if morphological variation was highly structured (*Herrel et al., 2008*). However, this does not appear to be the case within the morphospace of Greater Antillean *Anolis*. Some ecomorphs were primarily distinguished by the shape of the pelvis, and others by the shape of the pectoral girdle or limb length. Such low concordance in ecomorph 'signature traits' would hardly be possible if different parts of the locomotor skeleton were highly constrained to vary together. Thus, we anticipate that genetic variants that affect one part of the locomotor skeleton (e.g., hindlimbs) will not have systemic effects on the morphology of other parts (e.g., pelvis).

Even without such 'genetic constraints', morphological variation can be structured since individual responses to the environment make some phenotypes common and others rare. It has been hypothesized that the responsiveness of bone to mechanical stress imposed by the microhabitat contributed to the repeated evolution of ecomorphs – and in particular limb morphology – by consistently biasing the distribution of phenotypes that can be selected (*Losos et al., 2000*; *West-Eberhard, 2003*). Our experimental and comparative data provide three strong pieces of evidence against this hypothesis when applied to the locomotor skeleton. Firstly, plastic responses differed qualitatively between the two species. Although the plasticity of the ancestor is unknown, the lack of a shared response between species suggests that the developmental bias induced by mechanical stress is too transient to bear responsibility for the repeated evolution of ecomorphs. Secondly, traits that were highly plastic in either of the species did not appear to be highly evolvable within *Anolis*, and the changes in morphology caused by plasticity were poorly aligned with morphological divergence between the corresponding ecomorphs. This non-alignment persisted if only highly plastic or highly evolutionary divergent traits were considered. In other words, there was no evidence that evolution would proceed primarily through modification of plastic responses. Thirdly, the morphological variation induced by the microhabitat largely failed to explain individual differences in locomotor performance in those habitats. That is, while plasticity affects the phenotypes that *can be* selected in a particular microhabitat, it is perhaps unlikely to affect which phenotypes that *are* selected. We therefore conclude that morphological plasticity at best played a minor role in the repeated evolution of *Anolis* ecomorphs.

## When will plasticity take the lead in adaptive evolution?

The literature on developmental plasticity and evolution has repeatedly emphasized how bone development orchestrates functional integration between skeletons, muscles and other parts of the body, partly by being responsive to mechanical stress imposed by the environment (*Losos et al., 2000*; *West-Eberhard, 2003*; *Standen et al., 2014*; *Levis et al., 2018*). The apparent lack of a relationship between plasticity and adaptive divergence of the *Anolis* locomotor skeleton may therefore seem surprising. However, while the developmental biology of bone makes morphologies highly evolvable, it does not follow that plastic responses and adaptive divergence will be well aligned.

To see why, consider that environmentally induced phenotypes are most likely to 'take the lead' in adaptive evolution when (i) plasticity has a directional and recurrent effect on the phenotype distribution in a given environment; (ii) this shift in phenotype distribution has consequences for which phenotypes are selected (i.e., the ones in the population with highest fitness), and (iii) the phenotypes that are selected are heritable. We suggest that these three conditions are particularly likely to

hold either when environmental perturbations are severe or when the environmental variables that change are those that developmental systems use as cues in adaptive plasticity.

Firstly, if the environmental perturbation is severe, developmental plasticity is crucial since it enables individuals to maintain some (but variable) degree of functionality (*West-Eberhard, 2003*). The individuals may not be very fit in absolute terms but, if the perturbation is recurrent or affect a large fraction of the population, selection may retain at least some of the induced variants. As a result, the adaptations that evolve will tend to resemble phenotypes that once were environmentally induced, perhaps resulting in phenotypes that are distinctively novel (*Moczek et al., 2011*). That the morphology of the shoulder girdle in Polypterus fish forced to move on land resembles the shoulder girdle of stem tetrapods is at least compatible with this scenario (*Standen et al., 2014*). Furthermore, classic laboratory experiments by Waddington on *Drosophila* (*Waddington, 1953*), and more recent work on heat-shocked tobacco hornworms (*Suzuki and Nijhout, 2006*) demonstrates that it is possible to adapt through selection on novel, environmentally induced traits.

Secondly, when organisms have evolved an adaptive response to environmental cues, most individuals in a population will respond in similar (yet variable) ways to an extreme environment. These phenotypic 'extrapolations' may be reasonably fit, especially if the cue and selective environment are highly correlated (*Kouvaris et al., 2017*). As a result, locally adapted phenotypes should resemble environmentally induced phenotypes, a pattern supported by a meta-analysis of reciprocal transplant experiments in plants (*Radersma et al., 2020*). The phenotype dimensions that are adaptively plastic may in fact have particularly high evolvability since theoretical and empirical studies suggest that those dimensions harbour particularly high levels of additive genetic variation (*Draghi and Whitlock, 2012*; *Noble et al., 2019*; *Brun-Usan et al., 2020*). Thus, all three conditions above are fulfilled, making plasticity appear to take the lead in adaptive evolution (*Kovaka, 2019*; *Uller et al., 2020*).

The present results suggest that neither of these scenarios are likely to apply to the repeated evolution of similar locomotor skeletons in *Anolis* ecomorphs. While some microhabitats may rarely be encountered in nature by a given species, our experiments show that the plastic responses to these microhabitats do not radically change what phenotypes are available to selection, or which ones are fit. That is, selection on genetic variation appears sufficient to cause the evolutionary divergence in the locomotor skeleton between ecomorphs. Our experiments also suggest that *Anolis* lizards have not evolved to exploit the responsiveness of bone growth as a mechanism for adjusting their morphology to the microhabitat in which they happen to be born. The hallmarks of such adaptive plasticity are precise developmental 'switches' (e.g., seasonal polyphenism in butterflies and other insects; *Brakefield, 1996*) or physiological mechanisms that ensure that individuals can adjust their phenotype adaptively and often reversibly (e.g., vertebrate stress responses). In comparison, the flexibility of bone growth in *Anolis* appears a poor mechanism for adjusting individual morphology to match the local environment. In fact, the developmental mechanisms that underlie species differences in limb length may act very early in development, before any mechanical stress imposed by locomotion (*Sanger et al., 2012*). If this is not peculiar to *Anolis*, environmentally induced variation in bone morphology may typically exercise a transient effect on the direction of adaptation to microhabitat. Thus, in general, comparisons of the relationship between plasticity and adaptive divergence over different time scales will be important to reveal the extent to which plasticity is consequential, and when it facilitates or hampers adaptation.

In conclusion, we reveal a consistent but moderate degree of morphological convergence in the locomotor skeleton of Greater Antillean *Anolis*. Our comparative and experimental analyses demonstrate that phenotypic plasticity is unlikely to have contributed to the repeated evolution of limb and girdle morphologies in *Anolis* ecomorphs. Instead, we suggest that the locomotor skeleton has evolved within a subset of possible morphologies that are highly accessible through genetic change, allowing repeatability of adaptive divergence independently of plasticity.

## Materials and methods

### Micro-CT scanning

Museum specimens were subjected to micro computed tomography (micro-CT) scanning using a GE phoenix v|tome|x m system (source voltage 100 kV; source current 200 µA; isometric voxel size 30–

130 µm) at the Nanoscale Facility of the University of Florida, US. Reconstructed image stacks (software GE phoenix datos|x CT) were further processed using VGStudio MAX software (version 3.2) by applying manual thresholding to extract surface models of skeletal structures. We applied a number of criteria for selecting specimens, including completeness of the skeleton, sexual maturity, absence of malformations and capture in the native range of the species.

## Quantification of morphology

Linear measurements were directly obtained using the VGStudio MAX software. For each lizard, we measured the maximum length of humerus, femur, ulna, tibia, and the individual phalangeal elements (including the claw) of the longest digit of both fore- and hindlimb (in mm to the closest 0.01 mm). This was achieved by placing one landmark each on the proximal and on the distal end of the bone and extracting the distance between these 2 points in 3D-space (*Figure 1B*). A small number (N = 46; 0.53%) of individual measurements were missing due to fractured bones, and we imputed these missing values using the 'pcaMethods' package (*Stacklies et al., 2007*) based on all linear measurements of all individuals.

To obtain linear measures of the long bone thicknesses, the midpoint between proximal and distal end of each bone was determined and the image plane perpendicular to the longitudinal bone axis was selected. In this virtual transverse cross-section of the long bones, total bone diameter and cortical thickness (calculated from the difference of total bone diameter and bone cavity, divided by two) were measured to the closest µm by averaging two measurements taken on orthogonal axes of the bone to adjust for the slight deviation of the shape from a perfectly symmetrical ring (*Figure 1B*). All linear measurements were collected for one side (left or right) of each lizard. These measurements of lengths and thicknesses (including thresholding raw imaging data) were highly repeatable (Pearson's product-moment correlation $r = 0.992$, p-value<0.001, N = 40).

To quantify the shape of pectoral and pelvic girdles, meshes of segmented structures were exported as stl format, which were converted into ply format using the software MeshLab (*Cignoni et al., 2008*; version 2016.12). For each pectoral and pelvic girdle, we placed 18 landmarks on informative anatomical features (*Supplementary file 1A*) using the R package 'geomorph' (*Adams et al., 2019*; version 3.1.3). Landmarks for capturing variation in the pectoral girdles were partially adapted from *Tinius and Russell, 2014* and for the pelvic girdle from *Tinius et al., 2018*. Landmarks were placed on the left side of both structures, except for a small number of structures that showed damage on the left side (pectoral girdles, N = 23, 4.84%; pelvic girdles, N = 12, 2.53%). In these cases, we landmarked the right sides of the structures and used the R package 'StereoMorph' (*Olsen and Westneat, 2015*) to mirror landmarks onto the left side. All measurements of bone length, bone thickness and the placing of landmarks were performed blindly with respect to treatment and ecomorph and by the same person.

To obtain Procrustes shape variables, the 18 landmarks recorded per girdle were each subjected to a generalized Procrustes analysis' that superimposed landmarks and the resulting X-, Y-, and Z-coordinates were extracted (*Rohlf, 1999*). A statistical feature of this procedure is that seven degrees of freedom are lost in the case of 3D-landmarks (*Kendall, 1984*). A possible operation to remove this redundancy in dimensions is to extract principal components from the Procrustes shape variables. However, when principal components instead of Procrustes shape variables are transformed to standard normal deviates (see section' Transformation of morphological dataset' below), their properties change and biological signal would therefore be lost. Ultimately, the seven redundant dimensions in the Procrustes shape variables have no impact on the downstream analyses (e.g., trajectory analyses) and the analyses can therefore proceed on Procrustes shape variables (*Rohlf, 1999*).

For assessing the repeatability of the landmarking procedure (including the thresholding of raw imaging data to extract mesh files), we re-extracted and re-landmarked a set of 10 pectoral and pelvic girdles. For each of these datasets, we calculated the intraclass correlation coefficient (i.e., repeatability) following the method proposed by *Arnqvist and Mårtensson, 1998*; (see also *Fruciano, 2016*). In brief, we performed a Procrustes one-way ANOVA (function 'procD.lm' in the R package 'geomorph') with individual as categorical variable and using a residual randomization procedure with 999 iterations. The resulting mean squares were used to assess the variance due to differences among individuals compared to the total variance. Using this method, we found that the

repeatability of the entire procedure of extracting morphologies and applying landmarks was 0.98 for the pectoral girdle, and 0.94 for the pelvic girdle (*Supplementary files 1J and 1K*).

The centroid size of the pelvic girdle was highly correlated with the centroid size of the pectoral girdle ($r$ = 0.99, p-value<0.001, N = 375), and with snout-vent length (SVL) measured in lab-reared lizards using a digital caliper (*A. carolinensis*: $r$ = 0.96, p-value<0.001, N = 45; *A. sagrei*: $r$ = 0.97, p-value<0.001, N = 71). Centroid sizes of the pectoral girdles are also known to be highly correlated with SVL ($r$ = 0.98, p-value<0.001) across iguanid species, including *Anolis* lizards (*Tinius, 2016*). We therefore used the centroid size of the pelvic girdle as a proxy for body size. To generate an estimate of relative bone thickness and relative limb length (e.g., *Losos et al., 2000*), all univariate variables were divided by the centroid size of the pelvic girdle. The resulting dataset capturing morphological variation in the locomotor skeleton contained 108 landmark-derived traits (each 18 Procrustes shape variables with X-, Y-, and Z-coordinates per pectoral and pelvic girdle), 15 traits capturing limb length, eight traits capturing long bone thickness and one trait capturing body size, totaling 132 traits. Specimens lacking landmark data for pectoral or pelvic girdles were excluded from the analyses. The full dataset used in all analyses can be found in source data file 2.

## Transformation of morphological dataset

The morphological dataset is comprised of Procrustes shape variables (*Rohlf, 1999*), linear measurements of bone length and thickness, and centroid size as a proxy for body size, and are therefore not on a commensurate scale. The statistical handling of combined data sets is a topical issue in the analyses of morphological data (e.g., *Adams and Collyer, 2019*; *Collyer et al., 2020*). To ensure that the analyses of this dataset return biologically meaningful and interpretable results, we followed the approach advocated by *Adams and Collyer, 2019*. Raw values were scaled such that each column (i.e., trait) was centred to zero and divided by the standard deviation (*Adams and Collyer, 2019*). This transformation to standard normal deviates was performed prior to all analyses since it generates a phenotypic space in which trajectory analyses (*Adams and Collyer, 2009*) can be performed (see also *Legendre and Legendre, 2012*). For a similar approach, see *Stuart et al., 2017*. One downside of this approach is that the results are not interpretable in terms of changes in morphological shape. However, the focus of this study is to address how parallel ecomorph divergences are on different islands, and how well plastic responses are aligned with these divergences, across the entire morphology of the locomotor skeleton.

## Species selection

Since species status of some *Anolis* taxa is under debate, we included all *Anolis* taxa that are currently (December 2019) recognized as 'species' by the Reptile Database (*Uetz, 2019*). We aimed at covering at least three male individuals per *Anolis* species that fulfill the following criteria: 1) being classified as one of the six main ecomorphs (trunk-ground, trunk-crown, crown-giant, twig, grass-bush and trunk) by either *Losos, 2009* or *Nicholson et al., 2012* and 2) having a geographic origin on one of the main islands of the Greater Antilles (Cuba, Jamaica, Hispaniola and Puerto Rico). Species that are currently inhabiting smaller islands (*A. smaragdinus*, *A. maynardi*, *A. longiceps*, *A. fairchildi* and *A. brunneus* on the Bahamas and *A. conspersus* on Grand Cayman) or the mainland (*A. carolinensis*), but are closely related to congeners on the nearest main island (*Poe et al., 2017*) were assigned to the respective main island. One species (*A. scriptus*) was excluded since its current distribution is on the Bahamas, but it is phylogenetically nested within a clade of Puerto Rican species (*Nicholson et al., 2012*; *Poe et al., 2017*). Two species (*A. porcus* and *A. chamaeleonides*) were excluded since their ecomorph status in unclear (classifications vary between trunk, twig, crown-giant and 'twig-giant'; *Beuttell and Losos, 1999*; *Herrel and Holanova, 2008*; *Losos, 2009*; *Nicholson et al., 2012*). One species (*A. marron*) is classified as trunk ecomorph by *Losos, 2009*, but trunk-crown by *Nicholson et al., 2012*. Based on the results of our principal component analysis, we recognize the high similarity of *A. marron* to trunk ecomorphs and therefore classify it as trunk. A list of specimens and the museum identification codes can be found in **source data file 1**.

## Assignment of sex

To exclude shape changes that are attributed to sexual dimorphism rather than species differences, we only included males in our analyses. This is necessary because the nature of the vector-based

analysis does not allow controlling for sex as is common practice, for example, in linear mixed models. While this prevents us from detecting sex differences in the alignment between plasticity and evolutionary divergence, the present study was designed to test for an effect of microhabitat (broad or narrow surfaces) on the locomotor skeleton that is not specific to one sex.

During the process of micro-CT scanning, sex was assigned to all individuals based on external morphology, and cross-checked with sex identity based on information provided in museum catalogues. When such assignment of sex was not possible or ambiguous, we used the shape of the pelvic girdle to corroborate sex identity. The rationale is that, due to reproductive functions that differ between males and females, sex differences should be most pronounced in the shape of the pelvic girdle. We used a linear discriminant analysis to classify individuals lacking sex assignment based on a training set of individual with sex unambiguously assigned. We only accepted sex assignments that had a posterior probability of $\geq$0.8. The method was first validated on a large dataset consisting of 687 pelvic shape variables of 218 *Anolis* species for which sex assignments were available, and which was arbitrarily split into training (60%) and test (40%) sets. We found that 94.40% of all specimens were correctly assigned to their sex class, and we thus deem this method appropriate for assigning sex to individuals with unknown sex identity.

The final dataset consisted of 259 individuals from 95 species from the Greater Antilles with a minimum of one, a maximum of 14, and a mean of 2.73 ($\pm$1.59 standard deviations) individuals per species. We used principal component analysis (R package 'stats') to visualize broad patterns of variation between all individuals in this dataset (*Figure 2A*). In addition, we confirmed these patterns by performing phylogenetic principal component analyses (*Revell, 2009*) on the level of species and by constructing 'phylomorphospaces' (both using the R package 'phytools'; *Revell, 2012*); version 0.6–99).

## Quantification of convergence

We quantified convergence in morphology across the Greater Antillean *Anolis* using the Wheatsheaf index (*Arbuckle et al., 2014*). In brief, the Wheatsheaf index is a metric of how similar a group of distantly related taxa is given their phylogenetic distance (large indices indicate high degrees of convergence). The R package 'windex' (*Arbuckle and Minter, 2015*; version 1.0) calculates a Wheatsheaf indexes for a focal group of taxa (i.e., ecomorphs) given a trait matrix and a phylogenetic tree, and by permuting the dataset estimates the significance levels of the observed convergence. Since scaled values were used as input in these analyses, the Wheatsheaf indices were calculated with the setting 'SE = FALSE'. The input tree for both analyses was a phylogeny constructed by *Poe et al., 2017* that was pruned to contain the 95 focal species. Six *Anolis* species (*A. chlorodius*, *A. eladioi*, *A. loysiana*, *A. pecuarius*, *A. peynadoi* and *A. viridius*) were missing from the phylogeny since they were recently split from other species (*Köhler and Blair Hedges, 2016*). According to the phylogenetic affinities described in *Köhler and Blair Hedges, 2016*, species were grafted onto the phylogenetic tree using the function 'bind.tip' in the R package 'phytools' (*Revell, 2012*).

## Quantification of parallelism across islands

To assess the degree of parallelism between pairs of ecomorphs on pairs of islands, we followed a phenotypic trajectory analysis approach (*Adams and Collyer, 2007*; *Adams and Collyer, 2009*; *Collyer et al., 2015*; see also *Stuart et al., 2017*). We conducted these analyses both on the full data set and separately for the two girdles, limb length and bone (cortical) thickness. The latter makes the results for the girdles directly comparable to standard analyses of geometric morphometric data. For a given pair of ecomorphs on a pair of islands, the phenotypic trajectories between ecomorphs are described by a vector in multivariate space. The vector for a given island consists of independent two-sample *t*-values calculated for each of the 132 traits between the two focal ecomorphs. Thus, the elements of each vector are *t*-values, and each vector has 132 elements. *T*-values are the differences in trait means, divided by the standard errors of the differences and therefore represent the difference is trait means expressed in units of standard error. Resulting vectors for ecomorph pairs have both a direction and length in morphospace and can therefore be compared between island pairs using two metrics: the angle $\Theta$ between two vectors describes the alignment in direction and $\Delta L$ describes the difference between the length of two vectors. Values for the angle $\Theta$

fall between 0 and 180°, while ΔL can take positive or negative values. The angle Θ and the ΔL between vector a and b were calculated as follows:

$$\text{angle}\,\Theta = \text{acos}\frac{\text{sum(a*b)}}{\sqrt{(\text{sum(a}^2))}*\sqrt{(\text{sum(b}^2))}} * \frac{180}{\pi}$$
$$\Delta L = \sqrt{(sum(a^2))} - \sqrt{(sum(b^2))}$$

The significance levels of observed angles Θ and ΔL values were assessed using a permutation approach (see *Stuart et al., 2017* for a similar approach). To simulate parallelism, islands of origins were permuted for each set of ecomorph/island comparisons and 999 'parallel' angles Θ and ΔL values were calculated. Based on this distribution of angles Θ and ΔL values, we assessed if the observed values are significantly different from parallel. Similarly, to assess if the angles Θ deviate significantly from orthogonal (90°), we bootstrapped the individuals for each ecomorph/island comparisons by drawing with replacement. We then assessed if this 'bootstrapped' distribution was significantly different from 90°, that is, if orthogonal direction between two vectors could be rejected. Note that vectors drawn at random are expected to have an average angle Θ of 90°. We performed these tests for 53 ecomorph/island comparisons resulting from 15 ecomorph pairs and 6 island pairs (three ecomorphs missing on two islands preclude several comparisons).

The results of this t-test based trajectory analysis were confirmed by an alternative analysis that calculates phenotypic change vectors based on differences in multivariate means using a linear model with randomized residuals in a permutation procedure (*Collyer and Adams, 2007*). This was done using the function 'trajectory.analysis' in the R package 'RRPP' (*Collyer and Adams, 2018*; version 0.4.3) with the formula 'traits ~ island*ecomorph' and 999 iterations. Importantly, this analysis was performed separately for landmark and univariate data and, for the sake of comparison, the t-test based analysis was also repeated for these two data subsets. The observed angles Θ between the t-test based and the phenotypic change vector based approaches were highly correlated (landmark data: $r = 0.984$, p-value<0.001; univariate data: $r = 0.956$, p-value<0.001). We did not apply the phenotypic change vector based approach to test whether ecomorph divergences are aligned with environmentally-induced changes since the groups are different in kind (the plasticity data are within species and ecomorph data between species; see below) and therefore not appropriate for this statistical model.

A corollary of applying a *t*-value based calculation of trajectories in morphospace is that a decomposition of the vector into its 132 elements allows us to assess which of the morphological traits are contributing most to the overall ecomorph differences. Thus, sorting the 132 individual *t*-values by absolute values provides a ranking of traits according to impact on ecomorph divergence. Furthermore, we used the function 'scores' in the R package 'outliers' (version 0.14) to single out traits that had significantly (outside the 95th percentile) higher or lower *t*-values compared to the distribution of all 132 traits. We used this method to identify traits that significantly distinguish the evolutionary divergence between ecomorph pairs (trunk-ground and trunk-crown on Cuba) and the most plastic traits for both *A. carolinensis* and *A. sagrei*.

## Locomotor-induced variation in morphology

Skeletons are malleable by physical activity and adjust based on the stress that bones experience (*Hall, 2005*). To quantify morphological variation induced by locomotor activity, we raised juveniles of two species in structurally different habitats that foster phenotypes specialized in climbing or running. We selected a trunk-ground species specialized in running (*A. sagrei*), and a trunk-crown species specialized in climbing (*A. carolinensis*). Both are derived from the Cuban clade, which allows us to compare trajectories between plastic responses and morphological divergence between the ecomorphs in this clade. The following information is a brief summary of the detailed description in *Feiner et al., 2020*.

Twenty-four female and eight male adult lizards of each of the two species, *A. sagrei* and *A. carolinensis*, were collected in Palm Coast, Florida in April 2016 and brought to the animal facility at Lund University. We housed all animals in 80 litre cages (Wham Crystal box with mesh on top, 590 × 390×415 mm), with one male and three females per cage. Males were swapped between cages twice during the course of the experiment to increase the spread of parentage among experimental animals. Cages were enriched with twigs, hiding areas, basking spots and a water bowl. Plastic cups with moist vermiculite were provided for oviposition. Adult lizards were kept at a light cycle of 12

L:12 D and given access to basking lights (60 W) for 10 hr per day and a UV light (EXO-TERRA 10.0 UVB fluorescent tube) for 6 hr per day. Mealworms and crickets were provided *ad libitum*. Eggs were collected every second day and incubated at 26°C in individual small plastic containers filled two-thirds with moist vermiculite (5:1 vermiculite:water volume ratio) and sealed with clingfilm.

Hatchlings (at most 24 hr after hatching) were alternately assigned to experimental treatments that consisted of 80 litre cages (see above) with different interior fittings. The narrow treatment consisted of an arrangement of six unpainted wooden dowels, three 0.4 cm and three 0.8 cm in diameter (*Figure 4—figure supplement 2A*). The broad treatment contained two unpainted wooden planks (4 × 8×50 cm; *Figure 4—figure supplement 2B*). Both dowels and planks were angled at approximately 40°. Each cage contained a water bowl, and light conditions were identical to the adult setup described above. Cage walls were coated with Fluon, which proved fully effective in restricting lizards to the experimental interior fittings. A group of six juveniles were housed in a single cage since prior observations indicated that grouping lizards encourages locomotor activity and the development of a natural behavioural repertoire. To minimize negative effects due to social hierarchies (e.g. 'runting'), we grouped lizards of roughly the same age together (±3 days). Juvenile lizards were fed daily with different size classes of crickets according to their gape size, and crickets were dusted with vitamins and calcium (Zoo Med Reptivite Reptile Vitamins with D3) twice a week to promote normal bone growth. The structural habitat in this setup reflects biologically relevant conditions (range of natural perch diameters of *A. carolinensis* juveniles and *A. sagrei* juveniles, 0.3 to 4.4 cm and 0.3 to >10.8 cm, resp.; *Schoener, 1968*). Although the diameter of perch sites was comparable to previous studies testing the effect of structural habitat on limb plasticity in *Anolis* lizards (*Losos et al., 2000*; *Kolbe and Losos, 2005*; *Langford et al., 2014*), our setup provided more total surface area and greater cage size than in previous studies, and treatment commenced immediately upon hatching.

Most lizards in our experiment reached sexual maturity at the age of five months, at which time growth typically begins to slow down (*O'Bryant and Wade, 2001*), and we therefore chose this as the end point of the experiment. Before lizards were sacrificed (through a blow to the back of the head followed by neck dislocation and destruction of the brain), we conducted behavioural experiments (see below). Immediately after lizards were sacrificed, sex was determined based on external features and confirmed by inspecting internal anatomy *post mortem*. Lizards were fixed in 4% paraformaldehyde and stored in 100% ethanol. Morphological data was extracted from micro-CT scans as described above.

We successfully raised 154 juveniles of *A. sagrei* (N = 80 broad treatment, N = 74 narrow treatment) and 89 of *A. carolinensis* (N = 43 broad treatment, N = 46 narrow treatment) to 5 months of age. Survival rate from hatching until the termination of the experiment was 90% for *A. sagrei* and 87% for *A. carolinensis*. In this paper, we restricted the data collection of *Anolis* species to male specimens to allow comparison to the comparative dataset, resulting in the following sample size: *A. sagrei* (N = 41 broad treatment, N = 34 narrow treatment) and *A. carolinensis* (N = 22 broad treatment, N = 23 narrow treatment).

## Behavioural and performance assays

To quantify the effect of the treatment on locomotion, we performed two trials that capture different functional aspects of lizards. First, we assessed perching behaviour, and second, we scored running and climbing performances.

We tested perching behaviour of *A. sagrei* and *A. carolinensis* by scoring the propensity of lizards to climb a challengingly narrow dowel. Locomotion on narrow dowels presents a challenge for *Anolis* lizards, with species not specialized in manoeuvring narrow surfaces frequently losing balance or even falling off (*Losos and Sinervo, 1989*). The rationale of this behavioural assay was to test whether lizards that were reared in the narrow treatment show greater confidence with perching and basking on narrow surfaces.

The experimental setup of the perching assay consisted of a wooden dowel (52 cm), gradually narrowing towards the tip (diameter: 2 cm at base, 0.5 mm at tip), that was placed at 45° angle with the tip 5 cm under a heat lamp. During a trial, lizards were placed at the base of the dowel and we recorded the position it would chose to bask. To increase the motivation of the lizards to move towards the narrow tip of the dowel, and thus closer to the heat lamp, the experiment was performed at 16°C, and lizards were acclimatized to this temperature 30 min prior to the trial.

Temperatures were measured at the tip of the dowel directly under the lamp (26°C) and at 5 (21°C), 10 (20°C), and 20 cm (18°C) distance from the tip of the dowel. This means that the lizards could only approach the preferred body temperature range by climbing to the very top of the dowel (preferred body temperature of both species is around 33°C; *Corn, 1971*). All experiments were performed in the early afternoon between 1 and 3 pm. Each lizard was tested three sequential times, with each trial lasting for 1 min. During this trial period, we recorded positions (i.e., the distance away from the tip of the dowel) where the lizard was resting for more than 5 s and calculated the average position as the mean of all positions. A subset of lizards was also tested at 4 months of age to determine individual consistency (*Feiner et al., 2020*).

To assess locomotor performance we followed previous studies (*Losos and Irschick, 1996*; *Kolbe et al., 2016*) and tested both species using two different setups, a narrow track consisting of a 0.8 cm diameter-wide dowel, and a flat track (15 cm broad). To avoid confusion with the broad and narrow rearing treatments, we refer to lizards moving on the narrow track as 'climbing' and lizards moving on the broad track as 'running'. Both tracks were made of unpainted wood (the same material as the lizards were raised on), 1.2 m in length, and oriented at 37° angle to encourage lizards to run rather than jump (*Losos and Irschick, 1996*).

All trials were conducted blindly with respect to the treatment group of the lizard, as a second person allocated lizards from both a broad and a narrow treatment cage into individual cages (150 × 150×200 cm) the evening before the locomotor performance trials. This procedure standardized the handling and ensured that lizards could be placed on the running track swiftly. All trials were conducted in the mornings between 10 am and noon at 27.0°C (SD 0.99°C) with 53% humidity (SD 0.07%). All lizards were tested on both the climbing and running tracks on consecutive days with the order of tracks randomized between groups of lizards. Each lizard was tested three times successively on the same track and we only used the fastest of the runs for each individual per track for the analysis. Lizards were filmed in dorsal view using a GoPro camera (Hero3+ Silver, California, USA) at 30 frames-per-second. The camera was arranged on a tripod so that the lens and the racetrack were in parallel planes to allow for accurate analysis of videos. By using ImageJ (*Schneider et al., 2012*), we calculated total race time (over 1.2 m) and maximum speed.

At the beginning of each trial, we placed the lizard at the start of the track (i.e., the first interval line) and encouraged it to run by gently tapping it on the base of its tail with a paintbrush. A subset of lizards was also tested at 4 months of age to determine the individual consistency of locomotor performance throughout ontogeny (*Feiner et al., 2020*).

All behavioural assays were statistically analysed using linear mixed-effect models fit by maximum likelihood in the R package 'lme4' (*Bates et al., 2015*). Degrees of freedom for mixed effects models were estimated using the Satterthwaite's approximation. The cage ID a lizard was reared in was included as random effect in all models to control for shared environmental effects. All models were fitted with the main effect of log-transformed SVL and the first and second principal components of the 'Plasticity' and the 'Evolutionary divergence' traits (see above). The rationale here is to test if the traits of the locomotor skeleton that are plastic or evolutionary labile predict locomotor performance. For further details on the effects of the experimental treatment on locomotor performance, see *Feiner et al., 2020*.

## Acknowledgements

We thank David Blackburn and Edward L Stanley for providing us with access to the Nanoscale Research Facility (University of Florida, US) where all micro-CT scanning was conducted. We are grateful to the curators of museum collections who assisted with data collection: David Kizirian (AMNH), Lauren Scheinberg (CAS), Stephen Rogers (CM), Alan Resetar (FMNH), Richard Glor and Luke Welton (KU), Neftali Camacho (LACM), Jose Rosado and Jonathan B Losos (MCZ), Jim Mcguire and Carol Spencer (MVZ), Lee Fitzgerald and Toby Hibbitts (TCWC), Greg Schneider (UMMZ), Esther Langan, Kevin de Queiroz and Robert Wilson (USNM), and a special thanks to Coleman M Sheehy III and Leroy Nunez (UF) for coordinating museum loans. We thank Dan Warner and Tim Mitchell for assistance with collecting animals in the field, Hanna Laakkonen for help with lizard husbandry, assistance in the performance of experiments, Xi Yang for help with scoring locomotor performance data and Rachel Blow, Cédric Aumont and Maarten Vervoort for help with processing micro-CT data. We are also grateful to Dean Adams for statistical advice and to three anonymous reviewers for their

comments on the manuscript. This research was supported by a grant from the John Templeton Foundation (60501) to TU, a grant from the Royal Physiographic Society of Lund to NF, a Wenner-Gren postdoctoral fellowship to NF, and a Wallenberg Academy Fellowship from the Knut and Alice Wallenberg to TU. The study was conducted according to the Lund University Local Ethical Review Process under the permit number Dnr M 31–16.

## Additional information

### Funding

| Funder | Grant reference number | Author |
|---|---|---|
| John Templeton Foundation | 60501 | Tobias Uller |
| Royal Physiographic Society in Lund | | Nathalie Feiner |
| The Wenner-Gren Foundation | Postdoctoral Fellowship | Nathalie Feiner |
| Knut and Alice Wallenberg Foundation | Wallenberg Academy Fellowship | Tobias Uller |

The funders had no role in study design, data collection and interpretation, or the decision to submit the work for publication.

### Author contributions

Nathalie Feiner, Conceptualization, Data curation, Formal analysis, Investigation, Visualization, Methodology, Writing - original draft, Writing - review and editing; Illiam SC Jackson, Conceptualization, Data curation, Investigation, Methodology, Writing - review and editing; Kirke L Munch, Data curation, Formal analysis, Investigation, Methodology, Writing - review and editing; Reinder Radersma, Formal analysis, Investigation, Writing - review and editing; Tobias Uller, Conceptualization, Formal analysis, Supervision, Funding acquisition, Investigation, Writing - original draft, Project administration, Writing - review and editing

### Author ORCIDs

Nathalie Feiner (iD) https://orcid.org/0000-0003-4648-6950
Illiam SC Jackson (iD) http://orcid.org/0000-0002-7948-2860
Reinder Radersma (iD) https://orcid.org/0000-0001-8186-6348
Tobias Uller (iD) https://orcid.org/0000-0003-1293-5842

### Ethics

Animal experimentation: The study was conducted according to the Lund University Local Ethical Review Process under the permit number Dnr M 31-16.

### Decision letter and Author response

Decision letter https://doi.org/10.7554/eLife.57468.sa1
Author response https://doi.org/10.7554/eLife.57468.sa2

## Additional files

### Supplementary files

- Source data 1. Data file including catalogue IDs of museum specimens.
- Source data 2. Data file including raw data used in the analyses.
- Supplementary file 1. Supplementary Figures and Tables.
- Transparent reporting form

## Data availability

Raw scans are available at Morphosource under the project ID P1059, title 'Anolis sp.' (see source data file 1 for individual DOIs). The morphological raw data is available in the source data file 2. Supplementary file 1 contains Extended Methods and supplementary tables.

The following dataset was generated:

| Author(s) | Year | Dataset title | Dataset URL | Database and Identifier |
|---|---|---|---|---|
| Feiner N | 2020 | Anolis sp. | https://www.morpho-source.org/Detail/ProjectDetail/Show/project_id/1059 | Anolis sp., P1059 |

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
