## [Decision Letter]

Thank you for submitting your article "Plasticity and evolutionary convergence in the locomotor skeleton of Greater Antillean Anolis lizards" for consideration by *eLife*. Your article has been reviewed by three peer reviewers, one of whom is a member of our Board of Reviewing Editors, and the evaluation has been overseen by Diethard Tautz as the Senior Editor. The following individuals involved in review of your submission have agreed to reveal their identity: Kathryn Kavanagh (Reviewer #2); Paul Brakefield (Reviewer #3).

The reviewers have discussed the reviews with one another and the Reviewing Editor has drafted this decision to help you prepare a revised submission.

Summary:

This study is an important advance providing empirical evidence that developmental plasticity is not biasing options for selection in this widely studied Anolis radiation. Despite this lack of facilitating plasticity, the anole species do tend to evolve similar phenotypes in similar habitats. The authors conclude that any plasticity is a transient effect and that parallelism does not depend on plasticity.

It is a strength of the study that it analysed the shape of pectoral and pelvic girdles and long bone thickness, rather than just length, to see if skeleton parts that responded to mechanical stress are more evolvable. The result that 'signature traits' that evolved to adapt to a particular habitat differed among ecomorph pairs suggested that the skeletal parts involved in adaptation are not in fact parallel. But it is unclear whether this is just noise in the system or a real signal of the lineage or habitat.

Essential revisions:

1) The conclusions depend very much on a procedure in morphometrics that appears to be problematic, namely the "Transformation of morphological dataset" described in the text and the supplement. This is not well justified with no proof or reference. Why should one collect landmark data, apply some sort of random standardisation to them and add/treat them with other linear variables? This adds further random signal to the landmark data prior to the Procrustes fit. Hence, there needs to be a clear justification of this procedure.

2) Instead one could do a 2 Block Partial Least Squares analysis to compare 2 sets of variables regardless of the type of data (e.g. Procrustes coordinates of the skull versus diet). This is routinely used in this type of studies in which one wishes to calculate the strength of association (i.e. covariation) between 2 sets of different variables (Rohlf and Corti, 2000) and is a general analysis in squamate studies (Adams and Rohlf, 2000). There is no a priori or some sort of rescaling of the data beforehand.

3) The study includes shape data versus different types of size data with the size data quite possibly being correlated among themselves, but there is no test for this. One would have expected analyses similar to the ones carried out in Dickson et al., 2017. For instance at P.6 paragraph starting with "To determine…by ANCOVA and PGLS".

[Editors' note: further revisions were suggested prior to acceptance, as described below.]

Thank you for submitting your revised article "Plasticity and evolutionary convergence in the locomotor skeleton of Greater Antillean Anolis lizards" for consideration by *eLife*. Your article has been re-reviewed by three peer reviewers, one of whom is a member of our Board of Reviewing Editors, and the evaluation has been overseen Diethard Tautz as the Senior Editor. The following individuals involved in review of your submission have agreed to reveal their identity: Kathryn Kavanagh (Reviewer #2); Paul Brakefield (Reviewer #3).

The reviewers have discussed the reviews with one another and the Reviewing Editor has drafted this decision to help you prepare a revised submission.

Summary:

There is overall a positive assessment of your revisions, but there remains the central problem around the statistics. We acknowledge that there is no way to capture all aspects of morphological change, and it seems a good attempt to create a new combined metric including shape measurements in addition to linear measurements. However, the statistical morphometrics specialist among the reviewers (reviewer #1) still raises substantial issues that are outlined in the review. To resolve this, we should like to ask you to work out in more detail of what you had only generally indicated in your response letter. You should properly separate the different types of measurements (size, shape), analyze them individually and not use transformations. Your conclusions will evidently be strengthened by having individual analyses of traits and perhaps can be interpreted more clearly as individual evolved characters that could be linked to functional and developmental processes. Also, it would be more comparable to previous studies. You may in addition want to include the combined metric, but you need to discuss the issues with this that were brought up by the statistical expert.

*Reviewer #1:*

The response of the reviewers regarding the transformation of the data has not removed my concern. As it stands, the morphological dataset (prior to the "transformation") includes Procrustes aligned landmark coordinates, centroid size and some linear measurements, that is, a mixture of shape variables (of which 7 are redundant because of the degrees of freedom lost during the Procrustes fit) and various size variables. As such, this is already difficult to expect to be an easily interpretable signal (in particular size and shape signals are confounded and their potential relationships are irretrievable).

The authors refer to Stuart et al., 2017 as a similar approach to the construction of a morphological dataset. This is indeed a similar approach and Stuart et al., 2017 likewise suffers from the same weaknesses (linear dependency of the shape variables and increased difficultness of biological interpretations).

Of greater concern is the transformation of the data suggested, which further deteriorates the biological meaningfulness the authors expect to extract from the data: Procrustes shape variables, for instance, are linked by the geometry of the shapes they describe. Centring to zero and dividing each of them by their standard deviation completely destroy the meaningful relationships among these shape variables.

In summary, this is not a path to obtain a "valid multivariate space" for the analyses envisioned. The multivariate space obtained combines incommensurate variables and therefore lacks meaningful metric and inner product, compromising the valid usage of the notion of angle central to the research question of the paper.

The authors report that they obtain similar results when working on subsets of the data. They add Figure 4—figure supplement 1 to support this. However, this is not informative since it is a document without its context. They did some separate angle analyses on each subset of their data. Angles in high-dimensional spaces, as they occur particularly in geometric morphometrics, can appear contrary to our intuition from two- or three-dimensional planes and spaces. Therefore, some caution is necessary when interpreting them. In a two-dimensional space, given one vector, there is only one direction that is perpendicular to it. In three dimensions, by contrast, there is a whole plane that is perpendicular to the first vector, and there is thus a greater number of different ways in which the vectors can have different directions. Therefore, even relatively large angles between vectors can suggest that the resemblance between them is unlikely to be of random origin.

This is what I would then suggest them to do: work and present results for internally coherent subsets of data (size data, shape data) allowing easier interpretations, additional visualisations (shape changes for instance), and avoiding questionable inferences from a mathematically ill-defined morphospace.

*Reviewer #2:*

The authors have satisfactorily responded to the reviews and updated their manuscript appropriately. Another reviewer should look at the statistical analysis as I'm not an expert. I look forward to seeing the paper published.

*Reviewer #3:*

The authors have done a good, indeed excellent, job in their response. I would welcome publication.

---

## [Author Response]

Summary:This study is an important advance providing empirical evidence that developmental plasticity is not biasing options for selection in this widely studied Anolis radiation. Despite this lack of facilitating plasticity, the anole species do tend to evolve similar phenotypes in similar habitats. The authors conclude that any plasticity is a transient effect and that parallelism does not depend on plasticity.It is a strength of the study that it analysed the shape of pectoral and pelvic girdles and long bone thickness, rather than just length, to see if skeleton parts that responded to mechanical stress are more evolvable. The result that 'signature traits' that evolved to adapt to a particular habitat differed among ecomorph pairs suggested that the skeletal parts involved in adaptation are not in fact parallel. But it is unclear whether this is just noise in the system or a real signal of the lineage or habitat.

We are grateful to the reviewers for their overall positive feedback and for recognizing the significance of our study. We respond to the criticism that has been raised point-by-point below, but we would like to emphasize here that the ‘signature traits’ have been identified using the Wheatsheaf index in an explicit phylogenetic framework. This approach implements a randomization procedure to specifically test if the observed convergence in traits is stronger than expected by chance. We therefore maintain that the identified ‘signature traits’ are significantly different from noise and are therefore genuine signatures of convergent evolution between ecomorphs. To highlight this point in the manuscript, we modified the Results section:

“We adopted a methodology using a simple metric of convergence in a phylogenetic framework, the Wheatsheaf index (Arbuckle et al. 2014), to assess if levels of convergence for each ecomorph are higher than expected by chance given the phylogenetic relatedness.”

“However, convergence was greater than expected by chance for some traits […]”

Essential revisions:1) The conclusions depend very much on a procedure in morphometrics that appears to be problematic, namely the "Transformation of morphological dataset" described in the text and the supplement. This is not well justified with no proof or reference. Why should one collect landmark data, apply some sort of random standardisation to them and add/treat them with other linear variables? This adds further random signal to the landmark data prior to the Procrustes fit. Hence, there needs to be a clear justification of this procedure.

We apologize for not explaining sufficiently our methodology and its rationale. We followed the logic of phenotypic trajectory analyses (Adams and Collyer, 2009), applying a vector-based analysis on a highly multivariate dataset. It may be helpful to recognize that our approach is effectively the same as a recent analysis of parallel evolution (Stuart et al., 2017). In brief, this study quantified the alignment in morphological divergence between pairs of lake-stream sticklebacks in a multivariate space consisting of a mixture of landmark-based data as well as univariate measurements. Our analyses of island pairs of ecomorphs are thus equivalent.

Transformation of a multivariate dataset is essential for constructing a valid multivariate space in which the vector analyses can be performed (Adams and Collyer, 2019). Since our data includes both landmarks and univariate measurements, the transformation procedure precludes analyses that are typically the target of geometric morphometrics (e.g., quantification of shape changes associated with diet). However, transformation is crucial for our key question: whether or not evolutionary divergences are aligned with plastic responses. The transformation is not ‘random’, but scales all traits to a mean of zero and a standard deviation of one. The transformation is applied to Procrustes shape variables, rather than landmark coordinates, and the statement that this ‘adds further random signal to the landmark data prior to Procrustes fit’ is therefore not true.

In the original version of the manuscript, we failed to clearly point out that our procedure is necessary and follows that of Stuart et al., 2017, and we have therefore added this information to the Results section and Materials and methods section.

Results section: “This resulted in a morphological dataset consisting of 132 features or traits that were scaled to allow vector-based analyses on combined shape and univariate data following a similar approach by Stuart et al., (2017; see Materials and methods section).”

Materials and methods section: “This transformation was performed prior to all analyses and follows the approach described by Stuart, et al., (2017) to enable vector-based analyses comparing alignment between ecomorph pairs on different islands and alignment between plastic responses to microhabitat and evolutionary divergence between the corresponding ecomorphs.”

Importantly, the fact that we arrive at highly similar conclusions when we consider subsets of the data (e.g. only limb lengths, only bone thickness or only shape of girdles; Figure 4—figure supplement 1; Results section) affirms that the results are robust against the transformation procedure. In addition, to fully demonstrate that our vector analysis is robust, we applied an independent method (trajectory analyses using the R package ‘RRPP’; Collyer and Adams, 2007 and Collyer and Adams, 2018) separately for landmark and univariate data. While this approach has limitations in that only parallelism of vectors, but not orthogonality, can be statistically assessed, it delivered the same results: ecomorph divergences between islands are better aligned than random, but rarely indistinguishable. These additional results have been added to the manuscript:

Results section: “The results were consistent when pectoral girdle, pelvic girdle, limb length and bone thickness were analysed separately (Figure 4—figure supplement 1), and with an alternative method that relies on comparisons of multivariate means instead of t-tests (Collyer and Adams, 2007).”

Materials and methods section: “The results of this t-test based trajectory analysis were confirmed by an alternative analysis that calculates phenotypic change vectors based on differences in multivariate means using a linear model with randomized residuals in a permutation procedure (Collyer and Adams, 2007). […] We did not apply the phenotypic change vector based approach to test whether ecomorph divergences are aligned with environmentally-induced changes since the groups are different in kind (the plasticity data are within species and ecomorph data between species; see below) and therefore not appropriate for this statistical model.”

2) Instead one could do a 2 Block Partial Least Squares analysis to compare 2 sets of variables regardless of the type of data (e.g. Procrustes coordinates of the skull versus diet). This is routinely used in this type of studies in which one wishes to calculate the strength of association (i.e. covariation) between 2 sets of different variables (Rohlf and Corti, 2000) and is a general analysis in squamate studies (Adams and Rohlf, 2000). There is no a priori or some sort of rescaling of the data beforehand.

As the reviewer points out, the two-block partial least squares is a method for exploring patterns of covariation between two sets of variables. In the context of geometrics morphometrics, the variables are often shape variables but, as correctly pointed out, the method can be applied to other types of data as well. While this makes the method very versatile, it remains restricted to the quantification of covariation. There are no doubt interesting questions one could ask about covariation in our data set. However, the key question of our study is of a different nature, and cannot be addressed using a two-block partial least squares analysis.

Whether or not evolutionary divergences are aligned with plastic responses is fundamentally about shifts in morphospace and this analyses therefore needs to compare alignment and length of vectors between multivariate means (see Figure 3 and response to the previous comment for details). It is unclear how a two-block partial least squares analysis would be applicable to our specific research question.

Nevertheless, we recognize the utility in adopting a more widely used methodology, and have therefore validated our findings using the function ‘trajectory.analysis’ in the RRPP package (see above), a method for phenotypic trajectory analyses that should be familiar to researchers in the field of geometric morphometrics (see Materials and methods section).

3) The study includes shape data versus different types of size data with the size data quite possibly being correlated among themselves, but there is no test for this. One would have expected analyses similar to the ones carried out in Dickson et al., 2017. For instance, at P.6 paragraph starting with "To determine…by ANCOVA and PGLS".

The reason why our methodology differs from the one mentioned by the reviewer here (e.g. ANCOVA and PGLS) is that our question is different. There is no doubt that individual bone elements and other aspects of the locomotor skeleton are phenotypically integrated, but this is not an assumption of the method that we are applying. As explained above, the appropriate test of alignment between responses in phenotypic space compares vectors between multivariate means, not covariances between sets of variables (corrected for size, see Materials and methods section).

[Editors' note: further revisions were suggested prior to acceptance, as described below.]

Summary:There is overall a positive assessment of your revisions, but there remains the central problem around the statistics. […] You may in addition want to include the combined metric, but you need to discuss the issues with this that were brought up by the statistical expert.

We are grateful to the reviewers for their overall positive evaluation of our revision. We recognize that our statistical approach may appear unorthodox to parts of the geometric morphometrics community. However, we emphasize that we have carefully considered this statistical issue and follow the best practice advocated by Prof Dean Adams, who has developed both geometric morphometrics and other multivariate data analyses, including the vector-based trajectory analysis adopted here. To maximize the usefulness of our article for researchers facing similar challenges, we have expanded the relevant sections. Since *eLife* cannot accommodate supplementary text, we added the additional detail (including further references) to the Materials and methods section.

Reviewer #1:The response of the reviewers regarding the transformation of the data has not removed my concern. As it stands, the morphological dataset (prior to the "transformation") includes Procrustes aligned landmark coordinates, centroid size and some linear measurements, that is, a mixture of shape variables (of which 7 are redundant because of the degrees of freedom lost during the Procrustes fit) and various size variables. As such, this is already difficult to expect to be an easily interpretable signal (in particular size and shape signals are confounded and their potential relationships are irretrievable).The authors refer to Stuart et al., 2017 as a similar approach to the construction of a morphological dataset. This is indeed a similar approach and Stuart et al., 2017 likewise suffers from the same weaknesses (linear dependency of the shape variables and increased difficultness of biological interpretations).

Although not a common practice in the field of geometric morphometrics, creating multivariate datasets by merging entities on different scales is a well-established statistical operation (see Adams and Collyer, 2019 and Legendre and Legendre, 2012). However, we acknowledge that the analyses of morphological (and other forms of) data that are not on commensurate scale can be unfamiliar, and that this is an area of active method development (e.g., see Adams and Collyer, 2019, Collyer et al., 2020). To assist others that face similar challenges, we therefore expanded our rationale and procedure in the Materials and methods section.

Materials and methods section: The morphological dataset is comprised of Procrustes shape variables (Rohlf, 1999), linear measurements of bone length and thickness, and centroid size as a proxy for body size, and are therefore not on a commensurate scale. The statistical handling of combined data sets is a topical issue in the analyses of morphological data (e.g., Adams and Collyer 2019; Collyer et al., 2020). To ensure that the analyses of this dataset return biologically meaningful and interpretable results, we followed the approach advocated by Adams and Collyer, (2019). Raw values were scaled such that each column (i.e., trait) was centered to zero and divided by the standard deviation (Adams and Collyer, 2019). This transformation to standard normal deviates was performed prior to all analyses since it generates a phenotypic space in which trajectory analyses (Adams and Collyer, 2009) can be performed (see also Legendre and Legendre, 2012). For a similar approach, see Stuart et al., (2017). One downside of this approach is that the results are not interpretable in terms of changes in morphological shape. However, the focus of this study is to address how parallel ecomorph divergences are on different islands, and how well plastic responses are aligned with these divergences, across the entire morphology of the locomotor skeleton.

References added:

Rohlf, 1999.

Collyer, Davis and Adams, 2020.

Regarding the lost degrees of freedom, the reviewer is correct that seven dimensions of the Procrustes shape variables of the pelvic and pectoral girdles are redundant. However, it is known that these redundant dimensions do not contribute to the analyses performed in our study (Rohlf, 1999). Relying on principal components would not be an option when the Procrustes shape variables are combined with length measurements, and this is explained in the Materials and methods section.

Materials and methods section: To obtain Procrustes shape variables, the 18 landmarks recorded per girdle were each subjected to a ‘generalized Procrustes analysis’ that superimposed landmarks and the resulting X-, Y-, and Z-coordinates were extracted (Rohlf, 1999). A statistical feature of this procedure is that seven degrees of freedom are lost in the case of 3D-landmarks (Kendall 1984). A possible operation to remove this redundancy in dimensions is to extract principal components from the Procrustes shape variables. However, when principal components instead of Procrustes shape variables are transformed to standard normal deviates (see section ’Transformation of morphological dataset’ below), their properties change and biological signal would therefore be lost. Ultimately, the seven redundant dimensions in the Procrustes shape variables have no impact on the downstream analyses (e.g., trajectory analyses) and the analyses can therefore proceed on Procrustes shape variables (Rohlf, 1999).

Reference added:

Kendall, 1984.

Of greater concern is the transformation of the data suggested, which further deteriorates the biological meaningfulness the authors expect to extract from the data: Procrustes shape variables, for instance, are linked by the geometry of the shapes they describe. Centring to zero and dividing each of them by their standard deviation completely destroy the meaningful relationships among these shape variables.In summary, this is not a path to obtain a "valid multivariate space" for the analyses envisioned. The multivariate space obtained combines incommensurate variables and therefore lacks meaningful metric and inner product, compromising the valid usage of the notion of angle central to the research question of the paper.

The reviewer is mistaken that the transformation of data is inappropriate. In fact, the transformation of the data to standard normal deviates is a necessary step in generating a multivariate phenotype space from variables of incommensurate units and scale. This is discussed in detail in Adams and Collyer, 2019. However, as the reviewer correctly points out, it is important to recognize that these data and the results cannot be interpreted in terms of shape; however, it is perfectly valid to interpret them in terms of morphology. We now provide further details on the rationale and considerations of this procedure in the Materials and methods section (see response above) and also emphasize this in the Results section:

Results section: “While this approach means that differences between, for example, ecomorphs are not interpretable in terms of shape, the data capture the overall morphology of the locomotor skeleton, which is the primary focus of this study. However, for completeness, we also report the main results for girdles and limbs analysed separately.”

The authors report that they obtain similar results when working on subsets of the data. They add Figure 4—figure supplement 1 to support this. However, this is not informative since it is a document without its context. They did some separate angle analyses on each subset of their data. Angles in high-dimensional spaces, as they occur particularly in geometric morphometrics, can appear contrary to our intuition from two- or three-dimensional planes and spaces. Therefore, some caution is necessary when interpreting them. In a two-dimensional space, given one vector, there is only one direction that is perpendicular to it. In three dimensions, by contrast, there is a whole plane that is perpendicular to the first vector, and there is thus a greater number of different ways in which the vectors can have different directions. Therefore, even relatively large angles between vectors can suggest that the resemblance between them is unlikely to be of random origin.

The rationale for showing the results of the trajectory analyses on subsets of the data (girdles, limb length and bone thickness) is to demonstrate that the observed parallelism is distributed across the locomotor skeleton, rather than being restricted to only girdles or only limbs. To clarify this in the article, we have included an explanation to the Results section (see response above) and added it also to the Materials and methods sections:

Materials and methods section: We conducted these analyses both on the full data set and separately for the two girdles, limb length and bone (cortical) thickness. The latter makes the results for the girdles directly comparable to standard analyses of geometric morphometric data.

Further, we have added more detailed explanations to the figure that presents the results of the trajectory analyses on subsets of the data (Figure 4—figure supplement 1). This assures that the figure is embedded in the right context and can be interpreted as a stand-alone result.

Figure 4 legend: Alignment of the morphological difference between pairs of ecomorphs on the four Greater Antillean islands separately for individual aspects of the locomotor apparatus. Frequency plots of all 53 quartet comparisons (pairs of ecomorphs on island pairs) shows that the vast majority of angles Θ are between 0° and 90°. (…) The four plots are directly comparable to the plot in Figure 4A.

With respect to vector-based analyses, we reiterate that this is a standard approach in analyses of morphology, including shape in geometric morphometrics (e.g., the ‘trajectory.analysis’ function in the R package RRPP; Collyer and Adams, 2018). The Materials and methods section provides further references that will help readers to explore this methodology.

Reference added:

Adams and Collyer, 2007.

This is what I would then suggest them to do: work and present results for internally coherent subsets of data (size data, shape data) allowing easier interpretations, additional visualisations (shape changes for instance), and avoiding questionable inferences from a mathematically ill-defined morphospace.

As detailed above, we maintain that analyzing the morphology of the locomotor skeleton in a single dataset is statistically sound and allows for a more comprehensive answer to our specific questions (alignment between ecomorph divergences and environmentally induced changes). Yet, we are sympathetic to the reviewer’s concern that an analysis on a combination of morphological features can be hard to interpret (a criticism valid for virtually any multivariate analysis). Indeed, the results certainly cannot be interpreted in terms of shape. We therefore agree with the reviewer that reporting the results of trajectory analyses for limbs and girdles separately adds value. As was evident from our Supplementary Material, this clearly shows that the observed parallelism is distributed across the locomotor skeleton, rather than being restricted to only girdles or only limbs. In this revised manuscript, we retain the separate analyses as a Supplementary Material, but have ensured that it is clear to the reader that they can find these results there, and explore them in more detail if they so wish. We refer to the changes to the manuscript described above.